# Evaluating the Inequality of Medical Service Accessibility Using Smart Card Data

**DOI:** 10.3390/ijerph18052711

**Published:** 2021-03-08

**Authors:** Xintao Liu, Ziwei Lin, Jianwei Huang, He Gao, Wenzhong Shi

**Affiliations:** 1Department of Land Surveying and Geo-Informatics, The Hong Kong Polytechnic University, Kowloon, Hong Kong; xintao.liu@polyu.edu.hk (X.L.); iris-he.gao@connect.polyu.hk (H.G.); john.wz.shi@polyu.edu.hk (W.S.); 2Smart Cities Research Institute (SCRI), The Hong Kong Polytechnic University, Kowloon, Hong Kong; 3Institute of Space and Earth Information Science, The Chinese University of Hong Kong (CUHK), Shatin, Hong Kong; jianwei.huang@link.cuhk.edu.hk

**Keywords:** accessibility, medical service, inequality, smart card data

## Abstract

The measurement of medical service accessibility is typically based on driving or Euclidean distance. However, in most non-emergency cases, public transport is the travel mode used by the public to access medical services. Yet, there has been little evaluation of the public transport system-based inequality of medical service accessibility. This work uses massive real smart card data (SCD) and an improved potential model to estimate the public transport-based medical service accessibility in Beijing, China. These real SCD data are used to calculate travel costs in terms of time and distance, and medical service accessibility is estimated using an improved potential model. The spatiotemporal variations and patterns of medical service accessibility are explored, and the results show that it is unevenly spatiotemporally distributed across the study area. For example, medical service accessibility in urban areas is higher than that in suburban areas, accessibility during peak periods is higher than that during off-peak periods, and accessibility on weekends is generally higher than that on weekdays. To explore the association of medical service accessibility with socio-economic factors, the relationship between accessibility and house price is investigated via a spatial econometric analysis. The results show that, at a global level, house price is positively correlated with medical service accessibility. In particular, the medical service accessibility of a higher-priced spatial housing unit is lower than that of its neighboring spatial units, owing to the positive spatial spillover effect of house price. This work sheds new light on the inequality of medical service accessibility from the perspective of public transport, which may benefit urban policymakers and planners.

## 1. Introduction

Medical services are an integral part of urban public service systems. Access to medical services plays an important role in supporting public health and well-being for the development of sustainable cities. Furthermore, the adequacy and spatial distribution of medical resources directly affects the physical health of urban residents. As urbanization accelerates, urban extent increases and urban infrastructures expand to the periphery of a city, and thus the living space of people continues to spread to the surrounding areas. This is particularly true in a first-tier city such as Beijing, which has a large population. Consequently, it is becoming increasingly important to ensure that there is an even spatial distribution of urban facilities, including medical facilities, in large cities to provide the public with equal accessibility to these services. To this end, the Chinese government has made efforts to rationalize the allocation of medical resources, with the aim of providing equal access to medical services for the public, including vulnerable groups and low-income groups. This has included the implementation of policies on medical price supervision, all medical security recommendations, and diagnosis and treatment systems [1]. Beijing is a megacity, with a large population and hospitals. Despite the great efforts dedicated to providing equal medical accessibility, the high house price and the imbalanced distribution of the healthcare resources in Beijing could make the inequality of medical accessibility very serious in the city.

Many studies have evaluated the accessibility of medical services [2,3,4]. However, lacking data on transportation networks or the computational power of geographical information systems, most previous works calculated the travel costs (distance) to medical facilities using Euclidean or Manhattan distance based on vector maps [5]. However, public transport is the most common travel mode used by people in large cities, such as Beijing and Hong Kong. It is also normal and reasonable that, in most non-emergency situations, people who live in large cities access medical facilities by public transport. It would, therefore, be more accurate to calculate medical service accessibility in terms of the travel costs of using public transport systems.

In recent years, the acquisition of big data has become easier and ubiquitous owing to advances in information technology. Various types of big data, such as mobile phone signaling data and Global Positioning System (GPS) trajectory data, have been widely used in various fields [6,7]. However, despite the increasing availability of smart card data (SCD), which comprise data of passenger tap-ins and tap-outs at subway or bus stations, there has been little evaluation of medical service accessibility in terms of public transport system use.

When considering public transport as a travel mode to access medical services, the following basic questions arise. How can public transport-based accessibility to medical services be quantified differently from previous studies based on driving or Euclidean distance? How is public transport-based accessibility to medical services distributed in space and time? Are there inequities in public transport-based accessibility to medical services? Finally, how is public transport-based accessibility to medical services associated with the house price?

To answer these questions, a case study is conducted of Beijing, China. A large volume of SCD were collected and are used to calculate the travel costs of public transport-based travel to medical service facilities, and an improved potential model is applied to evaluate medical service accessibility via the 6th Ring Road in Beijing. A comparison is made between the proposed method in this study and a traditional method in a previous study, and the spatiotemporal patterns of public transport-based medical accessibility are explored to determine, for example, when and where the accessibility is high or low, and why this is so. The spatiotemporal inequality in public transport-based medical accessibility is determined at a high resolution. House price in Beijing is used as the proxy of socioeconomic context, and its relationship with medical accessibility is evaluated from local and global perspectives using spatial econometric analysis. This study combines a public medical-accessibility evaluation model with real SCD to comprehensively consider the spatiotemporal dynamics of accessibility patterns. The findings may contribute to the development of sustainable transport systems to enhance medical service accessibility in smart cities.

The remainder of this paper is structured as follows. Section 2 reviews previous studies on the calculation of medical service accessibility from different perspectives. Section 3 presents the study area and the data processed in this work. Section 4 details the methods used in this study, such as spatial autocorrelation analysis and an improved potential model for the estimation of medical service accessibility. The results and a discussion are provided in Section 5, followed by conclusions in Section 6.

## 2. Literature Review

Accessibility is a difficult concept to define. According to Gould, accessibility is a “slippery notion … [as it is] one of those common terms that everyone uses until faced with the problem of defining and measuring it” [8]. Therefore, many studies have calculated accessibility from different perspectives. The gravity model is one of the common methods used, and was pioneered by Hansen [9], who defined accessibility as the size of the opportunity for interaction and applied this to urban resource planning. Weibull [10] used Hansen’s definition [9] to consider the supply and demand competition among consumers, and introduced a population scale factor to improve the original gravity model.

Since these early studies, a large body of literature has accumulated on the analysis of spatial accessibility, which considers the effect of supply and demand on accessibility on a point scale. This approach has gradually emerged as a spatially rational reference basis for the assessment of urban public facilities and equality of access to resources. The two-step floating catchment area (2SFCA) method and potential model are examples of this approach, and have become the main methods for researching spatial accessibility in the field of public services. 2SFCA allows comparisons to be made across different locations [11] and, according to Luo and Wang, “is a special case of a gravity model of spatial interaction that was developed to measure spatial accessibility to primary care physicians” [12,13]. Wang, Du [4] first proposed the concept of observed hospital accessibility (OHA) and used an enhanced 2SFCA (E2SFCA) method and taxi trip data to evaluate OHA. The E2SFCA and variable two-step flowing catchment area (V2SFCA) methods may both work well in this study. Nevertheless, the major objective of our study is to verify the usability of the real public transportation big data in evaluating the inequality of medical accessibility in space and time, in which the overall trend of the study results will not be influenced by using the improved potential models [2,14]. The potential model comprehensively considers spatial barriers (time, distance, and so on) and the needs of residents, and accurately reflects residents’ access to facility resources in research units of smaller spatial scales than those used in 2SFCA [15].

At present, the potential model is more commonly used than 2SFCA to evaluate the accessibility of medical services. Salze, Banos [16] recognized that the calculation of accessibility combined with travel behavior better reflects reality, and used the potential model to analyze differences in accessibility to food outlets in different areas of a city. Peng, Zhang [17] comprehensively considered the important role of a transportation network in logistics activities and improved the potential model to evaluate the radiation scope of a logistics park. Moreover, considering multiple transportation modes is more accurate to measure the comprehensive accessibility [18,19]. Nevertheless, our study aims to demonstrate the spatiotemporal variations and patterns of medical service accessibility that only consider public transport, which is important to explore the social well-being and inequity from a particular angle.

Notably, the potential model combines the spatial effects of supply and demand with their own gravitational scale, comprehensively evaluates accessibility, and can combine actual-travel big data with a travel friction coefficient, which is ideal for the analysis of smart transportation-card data. However, the existing improved potential model for accessibility analysis has several flaws. For example, the travel friction coefficient is largely estimated from previous research methods, which only consider Euclidean distance or travel time. Moreover, travel costs are estimated based on the shortest path or travel time from the road network, not based on real transportation data [15]. Furthermore, it does not account for the fact that accessibility generally has strong socio-economic characteristics, as different regions or groups have various levels of access to public service facilities.

Notably, researchers have found that low-income individuals who live in affordable housing have the lowest accessibility levels and they have the highest risk of social exclusion and isolation in disadvantaged spaces [20]. Mavoa, Witten [21] analyzed the spatial differences in medical accessibility between various groups in Turkey, and found that there was spatial inequality in accessibility to medical treatment for urban residents of different ages, incomes, immigration status, and medical enrolment status. Kawakami, Winkleby [22] compared the differences in living services and resource access between different levels of communities in Switzerland, and they found that low-deprivation neighborhoods had a significantly lower prevalence than high-deprivation neighborhoods. Other studies have explored the root causes of unbalanced access to medical services in China by collecting socioeconomic indicators and using geographic-weighted regression [1]. Tao and Shen [23] compared the difference in accessibility to medical treatment between the household registration population and the floating population in urban areas, and found that the former had better medical treatment accessibility than the latter. Guo, Chang [24] used Hong Kong as a research object to analyze whether the wealth of residential areas affected residents’ access to urban facilities and found that poorer areas had more opportunities to access certain services than wealthy areas, not less. Zhao, Liu [25] confirmed that the proportion of the floating population in a community is related to the accessibility of community medical care, and revealed the unequal layout of urban facilities. Finally, although some previous studies on urban facility accessibility have analyzed spatial inequality, they used conventional spatial models, which do not take spatial autocorrelation and spatial spillover effects into consideration, and thus may lead to biased estimates [26].

In this study, we analyze the inequality of medical service accessibility using real SCD, the improved potential model, and spatial econometrics analysis in medical service accessibility data are considered.

## 3. Study Area and Data Processing

### 3.1. Study Area and Spatial Unit

The built area within the 6th Ring Road in Beijing was selected as the study area (Figure 1a). Beijing, the capital of China, is a world-famous and modern international city. It is located in Hebei Province on the northern part of the North China Plain. It is adjacent to Tianjin in the east and the west is adjacent to Hebei. Beijing has 16 administrative districts and a total population of 21 million within an area of 16,410.54 km^2^.

A hexagonal study unit was used and the study grid area contains a total of 2745 units within the 6th Ring Road (Figure 1b). A hexagonal grid was selected because it covers an area with more regularly sized hexagonal cells than a raster grid. Moreover, hexagonal cells are closer in shape to circles than to rectangular cells, and suffer from less orientation bias and sampling bias from edge effects than other cell shapes [27]. Furthermore, the authors of [28] suggested that an appropriate geographic unit for delineating service areas should represent 10 minutes’ walking distance (e.g., 1 km) and areas that contain a bus stop or subway station. Thus, the spatial resolution of the study grid is 1 km × 1 km.

### 3.2. Datasets and Data Processing

Four types of data are used, namely (1) hospital service data, (2) SCD, (3) population data, and (4) house price data.

#### 3.2.1. Hospitals Service Data

By the end of 2016, there were 9773 medical institutions, 117,041 sickbeds, and 299,460 health workers in Beijing [29]. We collected the address details of 192 hospitals in the study area from Amap (www.amap.com, accessed on 7 March 2021) and then geocoded these addresses using the Amap application programming interface to obtain the coordinates of each hospital. To quantify the scale of hospitals, we obtained the number of sickbeds in each hospital from the Beijing Municipal Commission of Health and Family Planning Information Center [30] and used these data as a measure of hospital size. In China, hospitals are classified into three classes according to their ability to provide medical and related services (e.g., medical care, education, and academic research), namely, Class I (primary), Class II (secondary), and Class III (tertiary) [31]. The classes of hospitals and the number of sickbeds in these hospitals across the study area are shown in Figure 2a,b, respectively.

Figure 2 shows that, the higher the class of a hospital, the more sickbeds it contains. In the following subsection, we estimate medical accessibility using the number of sickbeds of these hospitals as an indicator of their medical service capability. The number of sickbeds in a hospital is a key factor that is used to estimate the level of a hospital, which is itself an important factor in the ability of a hospital to attract patients [3,32,33,34].

#### 3.2.2. SCD

SCD from Beijing are used to identify origin and destination matrices and calculate travel costs. The SC dataset consists of 30,620,366 bus and subway travel trips from 11 April to 17 April 2016 (seven consecutive days in an entire week) in Beijing. The dataset contains information on the coordinates and schedules of tap-in–tap-out stations, travel cost time, and travel line code (Table 1). In addition, to ensure data quality and optimize computing efficiency, all of the SC records were cleaned by removing empty records, or records of alighting at invalid stations or of trips with a travel cost time that was too short. In addition, some records have system errors or a travel period that is not within the operating time period, and these records were also removed. Furthermore, some buses do not stop at their exact parking location, causing the latitude and longitude of the boarding and alighting station data to shift. To amend such errors, we use the line code and alighting time to estimate the boarding and alighting bus stations. To protect the privacy of passengers, records within each half hour are merged into a single record, which is shown as NUM in Table 1.

In Beijing, in 2016, the subway system comprised of 18 lines and 278 stations, and the public bus service comprised 1574 lines and 42,024 stations, and almost all bus and subway stations were located within the 6th Ring Road in Beijing. Public (e.g., metro and bus) transport in Beijing accounted for 49.3% of total commuter transportation, and was the most common travel mode for people in Beijing [35]. The boarding and alighting locations for bus and subway stations are typically distributed along roads (Figure 3a,b), making the data suitable for examining the characteristics of hospital access behavior [4].

As mentioned in Section 3.1, the spatial unit grid ensured that all spatial units had one matched bus stop or subway station. To identify the origin and destination matrices of the trips (Figure 4), we created a ring buffer at a distance of 800 m, as multiple tests have shown that a buffer zone of this size results in almost all hospitals having one matched bus stop or subway station [36]. Therefore, alighting stations (red points) that were 0–800 m from hospital coordinates were identified as representing trips to a hospital. Boarding stations located within the area of a spatial unit were identified as trips departing that spatial unit. In addition, to analyze temporal variations in medical service accessibility, we considered three time periods on weekdays and on weekends, as characterized by the Beijing Traffic Management Bureau: a morning peak period (between 07:00 and 09:00), an off-peak period (between 09:30 and 16:30), and an evening peak period (17:00 and 20:00). The first period is the beginning of the peak period in the morning, when many office workers and students visit doctors before their workday or school day begins. During the second period, most of the people who visit doctors have no constraints on their time. The third period is the peak time in the evening, when most of the people who visit doctors have finished work or school [6].

After identifying the origin and destination of the trips during these time periods, we merged the trips that had the same records, and finally derived a total of 9,538,435 origin–destination trips from 11 April to 17 April 2016.

#### 3.2.3. Population and Housing Price Data

The population data of China were downloaded from WorldPop (https://www.worldpop.org/geodata/summary?id=5777, accessed on 7 March 2021), where the 2016 population is aggregated in grids of 100 m × 100 m. The population in each spatial unit (see Section 3.1) is obtained by performing a bilinear resampling method. Figure 5a shows the population distribution in each spatial unit in the study area. In general, the population in an area (e.g., hexagonal grid unit) indicates potential patients [4].

Several studies have shown that economic inequality may lead to inequality of access to medical services [20,24]. In addition, house prices in an area can reflect, to some extent, the economic level of the residents in an area. Thus, house price data were obtained from the website of “Lianjia” (http://bj.lianjia.com/, accessed on 7 March 2021), a well-known real estate company in China, which divides the study area into 8000 communities. We conduct a spatial join analysis of these data to assign a house price to each spatial unit in the study area (Figure 5b), as the house price of a spatial unit can be used to represent the potential income of residents within the unit.

## 4. Methodology

The main objectives of this study are to quantify real public-transport commuting data-based (i.e., SCD-based) accessibility to medical services, determine the spatiotemporal distribution of this public transport-based accessibility to medical services, prove that there is inequality of access to medical services, and understand the relationship between socio-economic factors and medical service accessibility. This section presents details on the methodology used in this study (see Figure 6). First, data processing was performed, as briefly introduced in Section 3.2. Second, we quantitatively calculated the medical service accessibility by fitting the improved potential model. Last, we evaluated the inequality of medical service accessibility by using spatial econometrics analyses.

### 4.1. Measuring Medical Accessibility Based on Transport Mode

The potential model, which was developed from the gravity model, is a classic model for studying spatial interactions between an economy and society. However, the traditional potential model only considers the supply side, and overlooks the competition between different demand points that share the same facility for limited resources. Thus, it may limit the accuracy of an accessibility calculation. That is, when facilities of the same size serve different numbers of people, the calculation will yield the same accessibility. To solve this problem, Guagliardo [37] considered a so-called population impact factor as the demand side to improve the traditional potential model. We use this improved potential model, which is formulated as follows:(1)$$A_{i}=\overset{n}{\underset{j=1}{\sum }}A_{ij}=\overset{n}{\underset{j=1}{\sum }}\frac{M_{j}}{D_{ij}^{\beta }V_{j}}$$
(2)$$V_{j}=\overset{m}{\underset{k=1}{\sum }}\frac{P_{k}}{D_{kj}^{\beta }}$$
where *A_i_* represents the sum of the potentials generated by demand points *i* from all of the supply locations in the system, *A_ij_* is the potential generated by the supply point *j* to the demand point *i* when the travel friction coefficient is *β*, *M_j_* represents the scale of resources provided by supply point *j*, *D^β^* is the travel resistance factor (distance or time) from point *i* to point *j* when the travel friction coefficient is *β*, *V_j_* represents the population impact factor of the supply point on the previous basis, *P_k_* is the number of service demand population of the demand point *k*, and *D_kj_* is the distance cost of the demand point *k* and the facility *j*. When the travel cost decreases, *β* decreases and *A_i_* increases accordingly [5].

*β* is estimated from travel costs (i.e., travel distance and time), which are derived based on the SCD as follows:(3)$$n_{ij}=\left\{\begin{array}{c} t_{ij}^{-\beta_{1}}\\ D_{ij}^{-\beta_{2}} \end{array}\right.$$
where *n_ij_* represents the number of population flows from demand point *i* to the supply point *j*, $t_{ij}$ represents the travel time between demand point *i* and supply point *j*, and *D_ij_* indicates the Euclidean distance from demand point *i* to supply point *j*. Incorporating the actual travel time into the travel cost generates an accessibility calculation result that is closer to the real-life situation.

### 4.2. Spatial Autocorrelation

According to Tobler [38], “all attribute values on a geographic surface are related to each other, but closer values are more strongly related than are more distant ones”. According to the first law of geography, geographical data may be non-independent and related to each other because of spatial interaction and spatial diffusion [1]. In addition, medical service accessibility may exhibit the features of spatial autocorrelation [3]. To further explore the inequality of medical service accessibility, spatial autocorrelation analyses of accessibilities are applied, based on the above calculations. We test the spatial autocorrelation based on Moran’s *I* [39] and generate a local indicators for spatial association (LISA) cluster map. The formula of Moran’s *I* is stated below:(4)$$I=\frac{n\sum_{i=1}^{n}\sum_{j=1}^{n}w_{ij}\left(x_{i}-\bar{x}\right)\left(x_{j}-\bar{x}\right)}{S_{0}\sum_{i=1}^{n}{\left(x_{i}-\bar{x}\right)}^{2}}$$
(5)$$S_{0}=\overset{n}{\underset{i=1}{\sum }}\overset{n}{\underset{j=1}{\sum }}w_{ij}$$
where $S_{0}$ is the standard deviation of the sample and $w_{ij}$ is the spatial weight matrix, which in this study is the adjacent distance matrix. The matrix is normalized such that its largest eigenvalue is 1. $x_{i}$ and $x_{j}$ represent the observed values (medical service accessibility) of areas *i* and j, respectively, and $\bar{x}$ is the mean of the observed values. The value of Moran’s *I* ranges between −1 and 1 [39]. If the value of Moran’s *I* is greater than 0, this indicates that there is a positive spatial autocorrelation; that is, a high value is adjacent to high values. If the value of Moran’s *I* is less than 0, there is a negative spatial autocorrelation; that is, a high value is adjacent to low values. If the value of Moran’s *I* is close to 0, the spatial spread of data is random, without any spatial autocorrelation. The *Z*-score is used to determine the standard deviation, where a score of <−1.65 or >+1.65 indicates a 90% confidence level, a score of <−1.96 or >+1.96 indicates a 95% confidence level, and a score of <−2.58 or >+2.58 represents a 99% confidence level [40]. The *Z*-score is the most frequently used statistical indicator for the test of significance of Moran’s *I* [41,42], and is calculated as follows:(6)$$Z=\frac{I-E\left(I\right)}{\sqrt{VAR\left(I\right)}}$$

### 4.3. Spatial Durbin Model

We also take into account that house prices in an area (i.e., the potential income of residents in an area) may be an essential factor affecting the inequality of medical accessibility [3,43]. In a normal linear regression, the coefficients may reflect how medical service accessibility would change when there are changes in house prices. The indirect effects are always zero in this case. However, in spatial autoregression, the effects of a change in house prices on medical service accessibility are spatially spread [26,44]. To show that medical service accessibility is associated with the socio-economic context of the built environment, we use a spatial regression model to analyze the relationship between house prices and accessibility during different time periods [45]. We test the spatial regression model based on the spatial Durbin model, which takes the following form:(7)$$Y_{it}=\alpha +\rho \sum_{j=1}^{n}W_{ij}Y_{it}+\beta X_{i}+\theta \sum_{j=1}^{n}W_{ij}X_{j}+\varepsilon_{i}$$
where *i* is the spatial units; *j* is a spatial unit adjacent to *i*, where *j* ≠ *I*; Y is the dependent variable, which is the discrete coefficient of medical service accessibility; α is the intercept term; $W_{ij}$ is a spatial weight matrix [46]; t indicates different time periods; and ρ is a spatial scalar parameter, which has three possible scenarios. When ρ=0, there is no endogenous spatial interaction; that is, medical service accessibility is not associated with the spatial relationship among house prices in each spatial unit. When ρ>0, the medical service accessibility in a spatial unit tends to match that of neighboring spatial units, and there is a positive spatial agglomeration. When ρ<0, the medical service accessibility shows a dispersed spatial-distribution pattern. $X_{j}$ is a k vector of explanatory factors, β is the coefficient of the explanatory variable, θ is the coefficient of the spatial endogenous interaction term $W_{ij}X_{j}$ of the explanatory variable, and $\varepsilon_{i}$ is the error term [47]. Based on Equation (9), the spatial Durbin model (SDM) is used as the base model, and is defined as follows:(8)$$lnAccessibility_{it}=\alpha +\rho \sum_{j=1}^{n}W_{ij}lnAccessibility_{it}+\beta lnHouse\;price_{i}\;+\theta \sum_{j=1}^{n}W_{ij}lnHouse\;price_{j}+\varepsilon_{i}$$

The marginal effect of a variable cannot be directly obtained from the coefficient of the estimation of regression of a spatial measurement, as the coefficient is part of the recursive computation of marginal effects. This a very complex computation, and the coefficients alone cannot describe the effect of this change in accessibility. Therefore, it is necessary to calculate the direct and indirect effects of the estimation model to quantify the spatial spillover effects. By rewriting the SDM with indirect and direct effects as
(9)$$Y_{it}={\left(I-\rho W_{ij}\right)}^{-1}\left(\beta X_{i}+\theta \sum_{j=1}^{n}W_{ij}X_{j}+\varepsilon_{i}\right)$$
where $\varepsilon_{i}$ is a rest term containing the intercept and the error terms, the matrix of partial derivatives of the expected value of $Y_{it}$ with respect to the k*th* explanatory variable of $X_{i}$ in unit 1 up to unit N in time *x* becomes the following:(10)$$\left(\frac{\partial y_{it}}{\partial X_{1k}},\frac{\partial y_{it}}{\partial X_{2k}},\cdots \frac{\partial y_{it}}{\partial X_{Nk}}\right)={\left[I-\rho W\right]}^{-1}\left(\begin{array}{ccc} \beta_{k} & \cdots & W_{ij}\theta_{k}\\ \cdots & \cdots & \cdots \\ W_{ij}\theta_{k} & \cdots & \theta_{k} \end{array}\right)={\left[I-\rho W\right]}^{-1}\left[\beta_{k}+W_{ij}\theta_{k}\right]$$

There are non-diagonal terms present owing to the exogenous parameter $\beta_{k}$, the endogenous spatial lag parameter (ρ), and the exogenous spatial lag parameter ($\theta_{k}$). The average value of the diagonal elements represents the direct effect, whereas the average value of the non-diagonal elements represents the indirect effect (spatial spillover effect), and the sum of the direct effect and the indirect effect is the total effect. Elhorst [48] proposed the use of the maximum likelihood (ML) method to estimate the static spatial model. In this study, we also use the ML method to estimate the model.

## 5. Results and Discussion

### 5.1. Attenuation Parameter Results of Travel Time and Euclidean Distance

Figure 7a,b report the statistical results for the relationship between the number of potential patients and travel time, and for the relationship between the number of potential patients and Euclidean distance, respectively. These figures show that increased travel time and travel distance reduce the attractiveness of a hospital to people. It is similar pattern between potential patients and travel time/distance. It can be seen that potential patients rarely choose hospitals with a travel time of more than 3 h or a distance of more than 30 km (approximately 1.5 h of driving). As mentioned in Section 4.1, we assume that, even within the service catchment of the hospital, increasing travel costs would reduce its attractiveness to people, and would in turn reduce the supply capacity of the hospital service. Thus, these visualization results preliminarily support our assumptions.

We now calculate the attenuation parameters based on the exponential function to further test our hypothesis. The results show that the $R^{2}$ values of travel time and Euclidean distance are 0.87 and 0.80, respectively, which indicates that there are good correlations between travel time and the number of potential patients, and between Euclidean distance and the number of potential patients, when these data are fitted by an exponential function.

### 5.2. Spatiotemporal Medical Service Accessibility

Using attenuation parameters, hospital attraction, and Euclidean distance based on a vector map between each population grid unit and hospitals, we estimate the medical service accessibility for each spatial unit. Figure 8 shows the static distribution of overall medical accessibility in the study area. To further compare the visualized temporal differences in the medical service accessibility in the study area, we use the quantile method to classify the value of medical service accessibility into ten levels, and then normalize them. A white spatial unit indicates the lowest value of medical service accessibility, and the deepest red spatial unit indicates the highest value of medical service accessibility. Overall, the medical accessibility of residents in spatial grid units varies greatly among different locations in the study area. The level of accessibility decreases as the distance from the city center increases, and the level of accessibility in the eastern area is higher than that in the western area.

Medical service accessibility is also analyzed by considering changes that occur throughout the day. There is considerable temporal variation in the attractiveness of hospitals because the desirability of attending one hospital from another spatial unit in the study area changes substantially throughout the day, which also affects the equality of accessibility to medical services. To further comprehend the spatiotemporally of medical service accessibility, we classify medical service accessibility into three time periods on weekdays and weekends (a morning peak (from 07:00 to 09:00), an off-peak period (from 09:30 to 16:30), and an evening peak (from 17:00 to 20:00)), as shown in Figure 9.

The visualization results show that the spatial accessibility of medical services at different times varies in temporal dimensions. Medical service accessibility on weekends is better than that on weekdays, and that during peak periods is better than that during off-peak periods. In addition, on weekdays and weekends, medical service accessibility is highest during the morning peak period, while the most inconvenient time for potential patients to visit a doctor is during the off-peak period. Furthermore, the dynamic distribution of medical service accessibility has similar characteristics, and grid units with the highest level of accessibility are mainly located within the most central area in the 6th Ring Road. In contrast, the medical service accessibility is poor during all time periods for those potential patients living in non-central areas within the study area. However, some areas, such as in the northeast and southeast of the study area, have medical service accessibilities that are quite different from those of their surrounding areas.

### 5.3. Inequality Evaluation of Medical Service Accessibility

Table 2 provides the Moran’s *I* results of our medical service accessibility data for different time periods. As shown, Moran’s *I* is statistically significant for both static and dynamic values of medical service accessibilities within the study area. The values of Moran’s *I* for the accessibility of the medical services are larger than 0 and fluctuate within the range of 0.096–0.191, and the *p*-value and *Z*-scores (see Table 2) show that the confidence level of spatial autocorrelation is 99%. These data support our assumption that there is an obvious positive spatial autocorrelation in medical service accessibility.

The Moran’s *I* value calculated for medical service accessibility in weekday off-peak periods is lower than the other values, indicating that, during these times, the spatial correlation of medical service accessibility is relatively weak. This may indicate that people randomly or actively choose their preferred hospitals during these times, unrestricted by working hours. In contrast, the weekend morning peak accessibility has the highest value of Moran’s *I* (0.191), which shows that, during this time, there is a significant correlation between residents and hospitals in a spatial unit, across the entire study area. Finally, some spatial units are in better locations, yet have worse accessibility to hospitals, especially in the weekend peak periods.

Figure 10 shows the LISA cluster maps developed from the spatial autocorrelation analysis. Red areas indicate high aggregation and blue areas indicate low aggregation. Our visualization results show that medical service accessibility displays high and low aggregation. From the results of different time periods, it can be seen that there are similar spatial distributions and trends for high- and low-value aggregation areas in the spatial autocorrelation analysis. Low-aggregation areas are distributed in suburban regions and high-aggregation areas are distributed in central regions in the study area, which indicates that there is an imbalanced distribution of medical resources and services across different areas. Notably, medical resources are extremely abundant in central regions, but scarce in suburban regions.

Table 3 reports the regression results for the relationship between medical service accessibility and house price during different time periods, generated by the SDM with the ML estimation and deviation correction. As mentioned in Section 4.2, the regression coefficient of spatial measurement is part of the recursive computation of marginal effects, which has no special explanatory significance. Therefore, we measure the spillover effects of marginal effects to explain the regression results in Table 3.

In Table 3 and Table 4, A indicates overall medical service accessibility, WDMPA is weekday morning peak accessibility, WDOPA is weekday off-peak accessibility, WDEPA is weekday evening peak accessibility, WEMPA is weekend morning peak accessibility, WEOPA is weekend off-peak accessibility, and WEEPA is weekend evening peak accessibility.

Table 4 presents the average effects of the SDM. Our statistical results show that, irrespective of whether medical services accessibilities are static or dynamic, house price has a negative direct effect on accessibility. This means that a high house price in a spatial unit decreases the level of medical service accessibility within that unit. In contrast, the indirect effect analysis shows that high house prices increase the level of medical service accessibility in neighboring spatial units. This latter result indicates that the medical service accessibility of a higher house-price spatial unit is lower than that of its neighboring spatial units owing to the positive spatial spillover effect of house price.

Our findings reveal that house price has a negative direct effect on hospital accessibility, but that spatial units neighboring a spatial unit with a high house price have high medical service accessibility owing to the positive spatial spillover effect. That is, poor areas (slums, urban villages, and so on) adjacent to high house-prices regions have high medical service accessibility. However, this high medical service accessibility does not radiate to low-house-price agglomeration regions. Moreover, high- and mid-high house-price regions benefit from high medical service accessibility owing to their spillover effects. Land finance is one of the key indicators for assessing the performance of local governments [49,50]. Under this circumstance, house price is somehow pushed by the central government in urban development, while those poor aggregation regions are no exception. Furthermore, the rise in house price is one of the yardsticks for the political achievements of local government officials, implying that local governments always seek to set up policies in boosting economic growth. In this context, the total effect of medical services accessibility will improve.

## 6. Conclusions

In this study, we evaluate the inequality of medical service accessibility in multiple time periods using real public transport commuting data and an improved potential model. House price is used as a proxy of socioeconomic context to systematically assess how this affects medical service accessibility. Our statistical and visualization results show that space, time, and housing prices all have significant effects on the inequality of medical service accessibility. However, the visualization results show that the level of medical service accessibility is poor at all time periods for potential patients living in non-central areas within the study area, which implies that location has the most significant effect on the inequality of medical service accessibility. In addition, we find that there are temporal trends in medical service accessibility for people living in central areas, with medical service accessibility on weekends being better than that on workdays, and that during peak periods being better than that during off-peak periods, which means that the desirability of attending one hospital from another spatial unit in the study area changes substantially throughout the day. Overall, there is considerable temporal variation in the attractiveness of hospitals, which also affects the equality of accessibility to medical services. Policymakers may thus need to re-evaluate the current transportation network to develop an optimal distribution of medical services in Beijing.

Moreover, our findings reveal that overall medical accessibility in Beijing is strongly spatially correlated with house price. Spatial regression analysis of medical service accessibility and house price shows that accessibility within a higher-house-price spatial unit is lower than that in its neighboring spatial units owing to the positive spatial spillover effect of house price. Overall, in the static regression results, people living in or nearby high-house-price areas have higher accessibility to medical services, but those who live in poor agglomeration regions may not even have medium-level accessibility to medical services, which is specifically notable when considering the spatiotemporal characteristics. This may have further exacerbated the inequality of medical services problem.

Therefore, policymakers should consider the distribution of medical services in cities thoroughly and adopt more suitable (sustainable) development models in managing the distribution of medical services. Briefly, local governments should carefully plan the distribution of medical services in poor agglomeration regions. Unfortunately, this study does not consider accessibility by private transport. Owing to data limitations, this paper is limited to an analysis of public transport accessibility.

## Figures and Tables

**Figure 1 ijerph-18-02711-f001:**
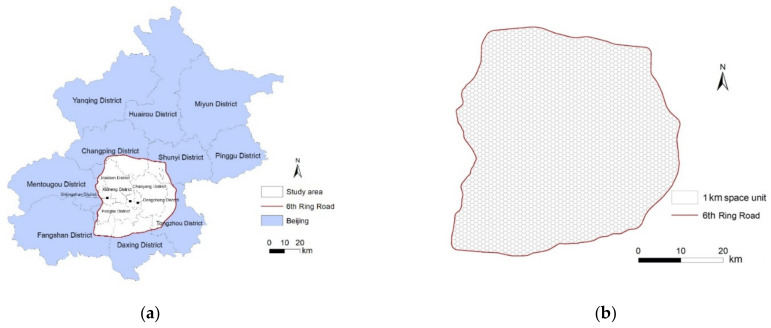
(**a**) Study area within the 6th Ring Road in Beijing, China; (**b**) expanded study area, showing its 2745 hexagonal (1 km × 1 km) spatial units.

**Figure 2 ijerph-18-02711-f002:**
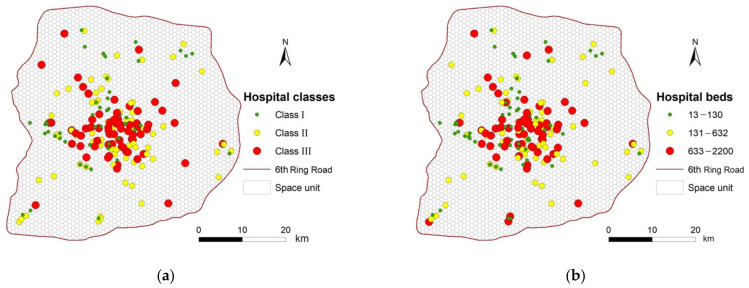
The spatial distribution of the 192 hospitals in the study area colored by (**a**) their class and (**b**) their number of sickbeds.

**Figure 3 ijerph-18-02711-f003:**
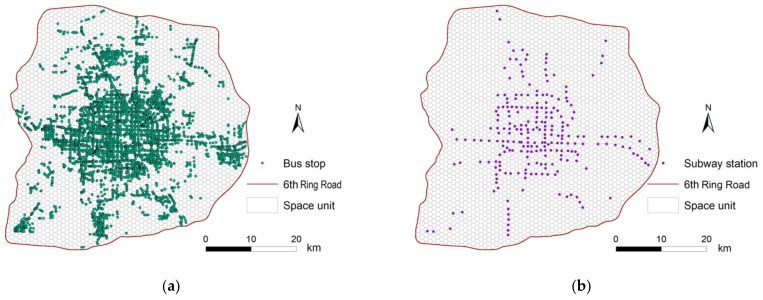
The distribution of public transport stations in Beijing: (**a**) bus stops; (**b**) subway stations.

**Figure 4 ijerph-18-02711-f004:**
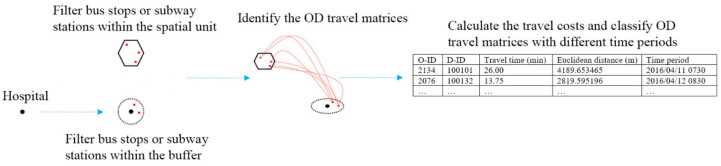
The schematic diagram of the derived origin and destination matrix (where OD = origin–destination, O–ID = origin identifier, D–ID = destination identifier).

**Figure 5 ijerph-18-02711-f005:**
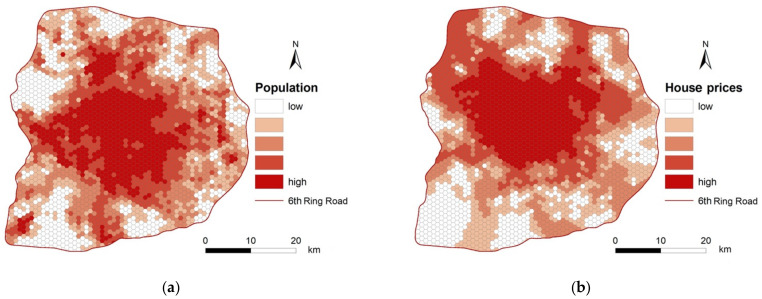
The spatial distribution of (**a**) population and (**b**) the house price of each spatial unit.

**Figure 6 ijerph-18-02711-f006:**
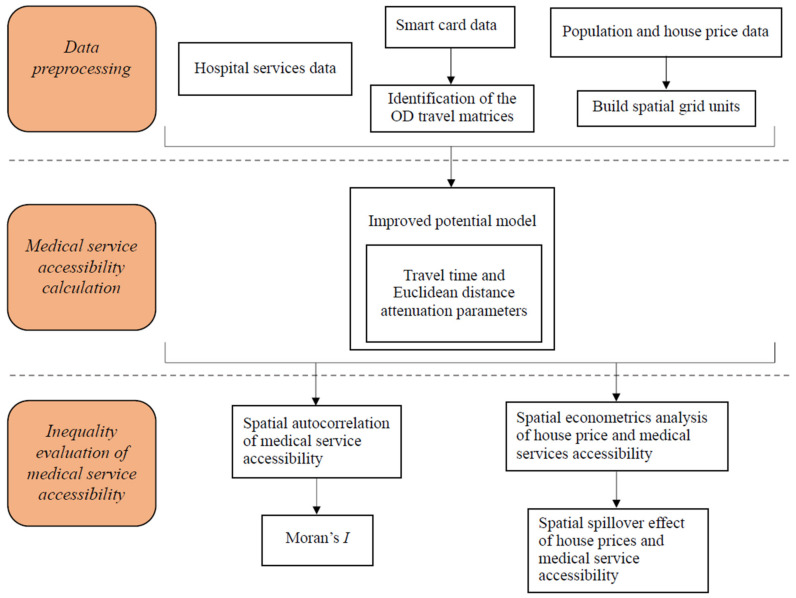
Workflow of the research.

**Figure 7 ijerph-18-02711-f007:**
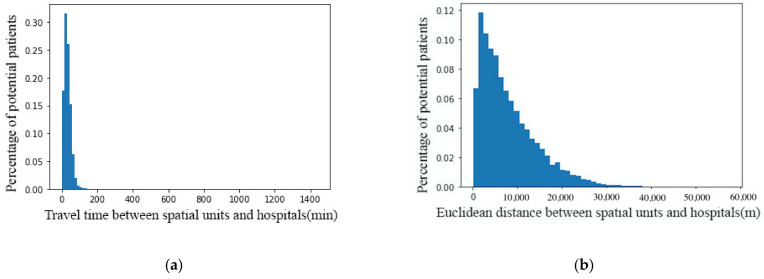
The relationship between potential patients and (**a**) travel costs and (**b**) travel time.

**Figure 8 ijerph-18-02711-f008:**
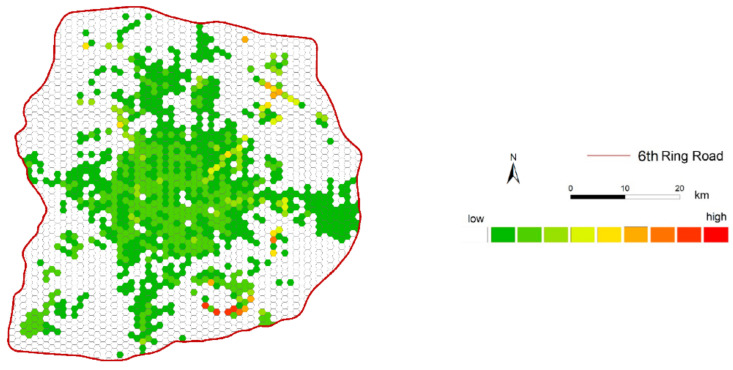
Distribution of medical accessibility in the study area.

**Figure 9 ijerph-18-02711-f009:**
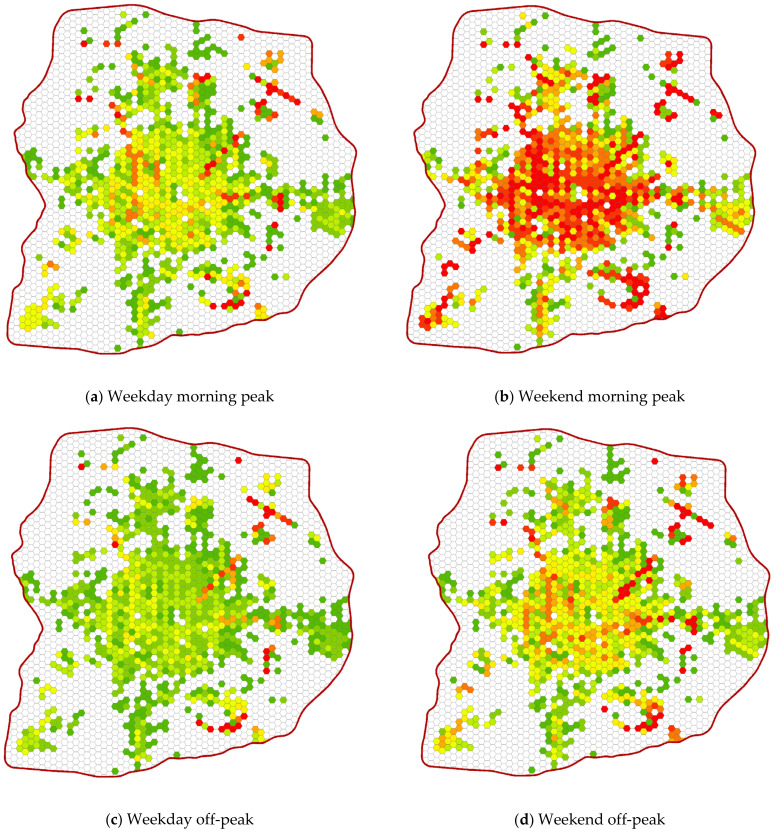
Medical service accessibility in different time periods, where (**a**,**c**,**e**) show the weekday dynamic accessibility and (**b**,**d**,**f**) show the weekend dynamic accessibility.

**Figure 10 ijerph-18-02711-f010:**
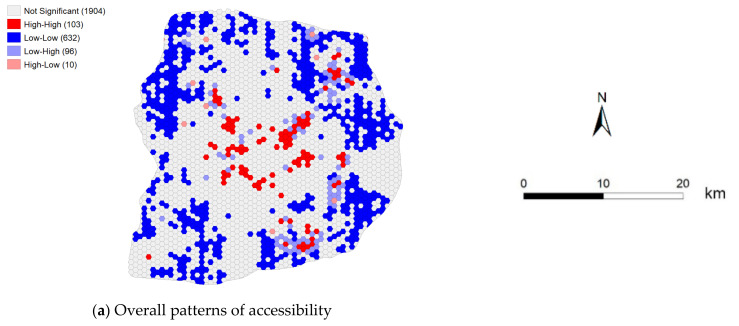
Local indicators for spatial association cluster maps of hospital accessibility. (**a**) Moran’s *I* of the overall patterns in accessibility; (**b**,**d**,**f**) Moran’s *I* of weekday accessibility in different time periods; and (**c**,**e**,**g**) Moran’s *I* of weekend accessibility in different periods.

**Table 1 ijerph-18-02711-t001:** Smart card data format.

Column Name	Meaning	Data Format
LINE_CODE	Bus/subway line number	1
ON_LON	Longitude of boarding station	116.225941
ON_LAT	Latitude of boarding station	39.915451
OFF_LON	Longitude of alighting station	116.231219
OFF_LAT	Latitude of alighting station	39.906253
ON_TIME	Boarding time	12 April 2016 0630
OFF_TIME	Alighting time	12 April 2016 0800
COST_TIME	Travel cost	99
NUM	Number of people traveling	3

**Table 2 ijerph-18-02711-t002:** Indicators related to the Moran’s *I* of hospitals.

Time Period	Accessibility	*Z*-Score	*p*-Value
Overall	0.187	13.4147	0.01 **
Weekday morning peak	0.136	14.3514	0.01 **
Weekday off-peak	0.096	5.9416	0.02 *
Weekday evening peak	0.135	15.6096	0.01 **
Weekend morning peak	0.191	17.9121	0.01 **
Weekend off-peak	0.136	13.2672	0.01 **
Weekend evening peak	0.135	17.3546	0.01 **

Standard errors are in parentheses, ** *p* < 0.05, * *p* < 0.1.

**Table 3 ijerph-18-02711-t003:** Spatial Durbin model of medical accessibility and house price. A, overall medical service accessibility; WDMPA, weekday morning peak accessibility; WDOPA, weekday off-peak accessibility; WDEPA, weekday evening peak accessibility; WEMPA, weekend morning peak accessibility; WEOPA, weekend off-peak accessibility; WEEPA, weekend evening peak accessibility.

Variables	(1)	(2)	(3)	(4)	(5)	(6)	(7)
	A	WDMPA	WDOPA	WDEPA	WEMPA	WEOPA	WEEPA
ln(house price)	−0.006 * (0.003)	−0.039 *** (0.012)	−0.018 ** (0.009)	−0.024 ** (0.010)	−0.073 *** (0.022)	−0.037 *** (0.014)	−0.051 ** (0.020)
W × ln(house price)	−0.009 *** (0.001)	−0.067 *** (0.004)	−0.026 *** (0.008)	−0.038 *** (0.004)	−0.121 *** (0.010)	−0.055 *** (0.005)	−0.063 *** (0.006)
Constant	0.09 *** (0.035)	0.675 *** (0.123)	0.265 *** (0.093)	0.374 *** (0.104)	1.189 *** (0.234)	0.563 *** (0.143)	0.655 *** (0.201)
Observations	2745						

Standard errors are in parentheses, *** *p* < 0.01, ** *p* < 0.05, * *p* < 0.1.

**Table 4 ijerph-18-02711-t004:** Average effects of the spatial Durbin model.

	A	WDMPA	WDOPA	WDEPA	WEMPA	WEOPA	WEEPA
Direct effect							
ln(house price)	−0.010 *** (0.0033)	−0.038 *** (0.012)	−0.018 ** (0.009)	−0.025 ** (0.010)	−0.074 *** (0.022)	−0.037 *** (0.014)	−0.053 ** (0.020)
Indirect effect							
ln(house price)	0.014 *** (0.004)	0.061 *** (0.014)	0.033 *** (0.012)	0.044 *** (0.013)	0.133 *** (0.029)	0.066 *** (0.018)	0.036 (0.047)
Total effect							
ln(house price)	0.004 *** (0.001)	0.022 *** (0.002)	0.014 *** (0.003)	0.020 *** (0.003)	0.059 *** (0.007)	0.029 *** (0.005)	−0.017 (0.049)

Standard errors are in parentheses, *** *p* < 0.01, ** *p* < 0.05.

## Data Availability

We make available all the results that are graphically represented with the map. Anyone on the internet with this link can view. https://drive.google.com/drive/folders/1R9VCl64Tbdgk6HKs9bFaNomCxhuKd2YI?usp=sharing (accessed on 28 February 2021).

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
