# Peer review of "Evaluating the Inequality of Medical Service Accessibility Using Smart Card Data"

_ijerph, 2021, doi:10.3390/ijerph18052711_

Round 1
Reviewer 1 Report
- The method you leveraged is a little outdated. Actually, 2SFCA method has been continuously improved. An enhanced two-step floating catchment area (E2SFCA) and Variable two-step floating catchment area (V2SFCA) methods would meet your original purpose to measure spatial accessibility.
Luo, W., & Qi, Y. (2009). An enhanced two-step floating catchment area (E2SFCA) method for measuring spatial accessibility to primary care physicians. Health & place, 15(4), 1100-1107.
Luo, W., & Whippo, T. (2012). Variable catchment sizes for the two-step floating catchment area (2SFCA) method. Health & place, 18(4), 789-795.
- All people do not take a public transportation. It depends on their preference. If you wish to measure spatial accessibility more accurately, you may also consider multiple transportation modes.
Mao, L., & Nekorchuk, D. (2013). Measuring spatial accessibility to healthcare for populations with multiple transportation modes. Health & place, 24, 115-122.
Dony, C. C., Delmelle, E. M., & Delmelle, E. C. (2015). Re-conceptualizing accessibility to parks in multi-modal cities: A Variable-width Floating Catchment Area (VFCA) method. Landscape and Urban Planning, 143, 90-99.
- reference [28] does not state that medical accessibility is related to house prices, which may be related to service fee. Actually, medical accessibility is highly associated with socioeconomic factors, such as income, education level, and so forth. You may wish to re-consider the variables to run spatial lag regression.
Kang, J. Y., Michels, A., Lyu, F., Wang, S., Agbodo, N., Freeman, V. L., & Wang, S. (2020). Rapidly measuring spatial accessibility of COVID-19 healthcare resources: a case study of Illinois, USA. International journal of health geographics, 19(1), 1-17.
- To measure the accessibility, you also may need to consider the operating hours of hospitals. I am not for sure whether you consider this. Otherwise, the accessibility measurements on weekend (or so) are not convincing.
Reviewer 2 Report
The paper is very well written. The research are important and interesting.
I only suggest to improve the conclusions which are too short in comparison with other parts of the paper.
FigurÄ™ 6 is very original and could be a part of conclusions as well.
Some very minor technical mistake appeared in the text: e.g. line 163 - should be km2 (not km2).
Reviewer 3 Report
The paper presents a research which is interesting from a methodological point of view, but I have some general concerns and doubts. The most important one regards results and conclusions.
I would definitely underline the relevance of your research.
To be clear:
- “These figures show that (line 381) increased travel time and travel distance reduce the attractiveness of a hospital to people”. I think is not so relevant result. In particular, because you are speaking about (line 383) “commute time of more than 3 h or a distance of more than 30 km (approximately 1.5 h of driving)”, which is incredibly high.
- On the contrary, (lines 411-413) “There is considerable temporal variation in the attractiveness of hospitals because the desirability of attending one hospital from another spatial unit in the study area changes substantially throughout the day, which also affects the equality of accessibility to medical services.” is a very interesting result.
- At the end your main conclusion is that “Overall, people living in or nearby high-house-price areas have higher ac-516 cessibility to medical services, but those who live in poor agglomeration regions may not 517 even have medium-level accessibility to medical services.”. I am not sure that this conclusion has a remarkable impact / significance in the contemporary academic literature.
What is very interesting in the paper that you present is the methodology. I would suggest to focus on describing that.
Moreover, the maps are tremendously clear and easily transmit the idea, but I would recommend that you make available all the results that are graphically represented with the maps. Appendices or links to online / cloud data could be a solution.
Furthermore, I suggest the authors to specify some details:
- Why evaluating just the medical service accessibility via the 6th Ring Road in Beijing?
- How could you take the SC data of Beijing?
Explaining both of these points in the paper could be very useful for not Chinese readers.
As last observation, I think that a proof-reading could help in the readability of the paper.
Reviewer 4 Report
Dear authors:
First of all, I would like to congratulate you on the work done. The topic addressed is relevant; the methodology with which the work was carried out is adequate. Furthermore, the contributions proposed in their conclusions are relevant to their field of knowledge.
Next, I propose some possibilities for improvement for the result of your paper.
The abstract has a correct content and reflects well the main idea of the paper. It is considered that the abstract could be improved if a sentence is included in which the research methodology is explained.
The keywords chosen for the paper are appropriate to the contents of the paper.
The introduction has a suitable structure and serves in a correct way to contextualize the content of the work. This content can be improved if some aspects are taken into account:
- Contextualize the geographical area in which the study is framed, determine the specific characteristics of this territory (lines 44 onwards).
- You detail the information of the previous studies that have been taken into account, and that are referred to in lines 47 etc ...
- Relate the research questions posed on line 69 with the design of the subsequent research.
The review of the scientific literature that supports the study is considered adequate, and allows a correct understanding of the scientific background of the research process that has been applied.
Section 3, which specifies the area of study and the procedure for using the data, is considered very appropriate, given the nature of the study presented. As far as possible, it is recommended to improve the quality of the figures provided in this section.
The explanation of the methodology section is considered adequate, precise and thorough
In the presentation of results, it is proposed to review if it is presented in the order consistent with the previous methodology presentation. Also, to the extent possible, improve the quality of the images.
On the other hand, it is strongly recommended to include a results discussion section with the results obtained.
The conclusions presented are considered correct. It would be better if practical considerations are added, which make the application of the conclusions reflected by the study visible to the problem it deals with.
The references section is considered adequate, although it should be enriched with the references that are included in the results discussion section.
I wish that these portings improve the final result of the paper
Round 2
Reviewer 1 Report
Thank you for your revision.
The remaining one is that it would be better to incorporate the administrative unit map with medical service accessibility measure maps. Figure 8, 9 and 10 seems not that helpful for effectively visualizing your results.
Thank you.
Reviewer 4 Report
Dear authors: I would like to congratulate you on the changes you have made in your paper. the end result has been improved thanks to this process.
Best regards
